# Hyperthermia as a Potential Cornerstone of Effective Multimodality Treatment with Radiotherapy, Cisplatin and PARP Inhibitor in *IDH1*-Mutated Cancer Cells

**DOI:** 10.3390/cancers14246228

**Published:** 2022-12-17

**Authors:** Mohammed Khurshed, Elia Prades-Sagarra, Sarah Saleh, Peter Sminia, Johanna W. Wilmink, Remco J. Molenaar, Hans Crezee, Cornelis J. F. van Noorden

**Affiliations:** 1Department of Medical Oncology, Cancer Center Amsterdam, Amsterdam UMC, University of Amsterdam, 1105 AZ Amsterdam, The Netherlands; 2Department of Medical Oncology, Cancer Center Amsterdam, Amsterdam UMC, Vrije Universiteit, 1081 HV Amsterdam, The Netherlands; 3Department of Radiation Oncology, Cancer Center Amsterdam, Amsterdam UMC, Vrije Universiteit, 1081 HV Amsterdam, The Netherlands; 4Department of Hematology, Cancer Center Amsterdam, Amsterdam UMC, Vrije Universiteit, 1081 HV Amsterdam, The Netherlands; 5Department of Radiation Oncology, Cancer Center Amsterdam, Amsterdam UMC, University of Amsterdam, 1105 AZ Amsterdam, The Netherlands; 6Department of Genetic Toxicology and Cancer Biology, National Institute of Biology, 1000 Ljubljana, Slovenia

**Keywords:** isocitrate dehydrogenase, PARP, hyperthermia, *D*-2-hydroxyglutarate, radiotherapy, cisplatin

## Abstract

**Simple Summary:**

Mutations in the isocitrate dehydrogenases 1 and 2 are causal in the development and progression of high-grade chondrosarcoma, high-grade glioma and intrahepatic cholangiocarcinoma. Due to the lack of effective treatment options, these aggressive types of cancer have a dismal outcome. Since hyperthermia increases the efficacy of DNA-damaging therapies such as radiotherapy and platinum-based chemotherapy, we introduce hyperthermia as the cornerstone of a multimodality treatment regimen for patients with *IDH1*^MUT^ solid cancer. These regimens include (I) hyperthermia added to conventional treatment with radiation and/or chemotherapy such as cisplatin and (II) hyperthermia in combination with PARP inhibitors.

**Abstract:**

Mutations in the isocitrate dehydrogenase 1 (*IDH1*^MUT^) gene occur in various types of malignancies, including ~60% of chondrosarcomas, ~30% of intrahepatic cholangiocarcinomas and >80% of low-grade gliomas. *IDH1*^MUT^ are causal in the development and progression of these types of cancer due to neomorphic production of the oncometabolite *D*-2-hydroxyglutarate (*D*-2HG). Intracellular accumulation of *D*-2HG has been implicated in suppressing homologous recombination and renders *IDH1*^MUT^ cancer cells sensitive to DNA-repair-inhibiting agents, such as poly-(adenosine 5′-diphosphate–ribose) polymerase inhibitors (PARPi). Hyperthermia increases the efficacy of DNA-damaging therapies such as radiotherapy and platinum-based chemotherapy, mainly by inhibition of DNA repair. In the current study, we investigated the additional effects of hyperthermia (42 °C for 1 h) in the treatment of *IDH1*^MUT^ HCT116 colon cancer cells and hyperthermia1080 chondrosarcoma cancer cells in combination with radiation, cisplatin and/or a PARPi on clonogenic cell survival, cell cycle distribution and the induction and repair of DNA double-strand breaks. We found that hyperthermia in combination with radiation or cisplatin induces an increase in double-strand breaks and cell death, up to 10-fold in *IDH1*^MUT^ cancer cells compared to *IDH1* wild-type cells. This vulnerability was abolished by the *IDH1*^MUT^ inhibitor AGI-5198 and was further increased by the PARPi. In conclusion, our study shows that *IDH1*^MUT^ cancer cells are sensitized to hyperthermia in combination with irradiation or cisplatin and a PARPi. Therefore, hyperthermia may be an efficacious sensitizer to cytotoxic therapies in tumors where the clinical application of hyperthermia is feasible, such as *IDH1*^MUT^ chondrosarcoma of the extremities.

## 1. Introduction

Mutations in the isocitrate dehydrogenase 1 gene are driving events in the development and progression of various types of cancer, including glioma, chondrosarcoma, cholangiocarcinoma and acute myeloid leukemia (AML) [1,2,3]. IDH1 is a homodimeric enzyme (*IDH1*^WT^) that catalyzes the conversion of isocitrate to α-ketoglutarate (αKG) with concomitant reduction of NADP^+^ to NADPH in the cytoplasm [1]. The heterozygous hotspot mutations in *IDH1* lead to the formation of *IDH1*^WT/MUT^ heterodimers (*IDH1*^MUT^) with a neomorphic IDH activity that converts αKG into the oncometabolite *D*-2-hydroxyglutarate (*D*-2HG) [4]. This activity induces a decrease in intracellular reducing power (NADPH) and an accumulation of *D*-2HG in *IDH1*^MUT^ cancer cells (Figure 1), resulting in improved responses to irradiation and chemotherapy in solid tumors [1,5].

Accumulation of *D*-2HG has been implicated in tumor progression through its inhibitory effects on αKG-dependent dioxygenases, which cause suppression of the homologous recombination repair system of DNA double-strand breaks (Figure 1) [6]. The absence of a proper functioning homologous recombination system leads to an increase in double-strand breaks and programmed cell death [7]. Homologous recombination deficiency also sensitizes *IDH1*^MUT^ cancer cells to DNA-repair-inhibiting agents such as poly-(adenosine 5′-diphosphate–ribose) polymerase inhibitors (PARPi); the PARP inhibition causes reduced repair of both double-strand breaks by homologous recombination deficiency and single-strand breaks (Figure 2) [8]. Recent reports demonstrated sensitivity of *IDH1*^MUT^ cancer cells to DNA damage and sensitization to PARPi in clinically relevant models, including patient-derived glioma and sarcoma cell lines as well as in vivo models [6,8,9,10,11,12].

Hyperthermia is an anti-cancer therapeutic strategy in which the tumor temperature is elevated to 40–43 °C for approximately 1 h. Hyperthermia has multiple anti-cancer effects, such as enhancing immune responses, inducing heat shock proteins and interfering with DNA metabolism [13,14,15]. When combined with other therapies, such as radiotherapy (RT) and chemotherapy, hyperthermia increases treatment responses in several types of cancer [16,17]. One of the important effects of hyperthermia is heat-induced degradation of proteins involved in DNA repair, which results in a deficient homologous recombination repair system and increased DNA damage [13,16]. Second, the accumulation of reactive oxygen species (ROS) and inhibition of antioxidant mechanisms caused by hyperthermia increase oxidative stress in cells [18,19]. Third, blood flow and tissue perfusion increase significantly due to hyperthermia, which improves tumor oxygenation and thereby the sensitivity of (initially hypoxic) cancer cells to RT, since well-oxygenated cancer cells are more sensitive to RT [20]. The hyperthermia-induced deficiency in homologous recombination DNA repair provided a strong rationale for adding PARP inhibition to the combination treatment of HT and RT and/or chemotherapy [21,22,23]. Hyperthermia-induced sensitivity of *IDH1*^MUT^ cancer cells is likely mediated by at least two components causing cell death: first, the altered metabolism and relatively low reducing power (NADPH) of *IDH1*^MUT^ cancer cells [20] and, second, the reduced effectiveness of DNA repair systems. Dysfunctional homologous recombination repair systems and altered oxidative stress responses caused by altered metabolism explain the susceptibility of *IDH1*^MUT^ cancer cells to the combinational treatment with hyperthermia and PARPis.

The first goal of the present study was to provide in vitro evidence of the effect of hyperthermia as an emulsifier in the treatment of patients with *IDH1*^MUT^ cancers, either in addition to conventional cytotoxic treatments (RT and/or chemotherapy) or in combination with conventional cytotoxic treatments and PARPis. This multimodality approach, which utilizes hyperthermia and PARPis of *IDH1*^MUT^ solid tumors, may provide a novel, therapeutic strategy for *IDH1*^MUT^ solid cancers that could locally disable homologous recombination and single-strand breaks repair, thereby sensitizing cancer cells to DNA-damaging agents and increasing cell death (Figure 2).

## 2. Materials and Methods

### 2.1. Cell Culture 

HCT116 *IDH1*^MUT^ knock-in colon carcinoma cells (*IDH1*^MUT^ HCT116 cells), generated by AAV-targeting technology GENESIS [24], were kindly provided by Horizon Discovery (Cambridge, United Kingdom). Hyperthermia1080 chondrosarcoma cells were gifted by Dr. Hamann (Department of Experimental Immunology, Amsterdam UMC, location AMC). *IDH1*^MUT^ and *IDH1*^WT^ HCT116 cells were cultured in McCoy’s 5A medium (Gibco; Life Technologies; Thermo Fisher Scientific, Waltham, MA, USA) in 5% CO_2_ at 37 °C. Hyperthermia 1080 chondrosarcoma cells were cultured in 10% CO_2_ at 37 °C in complete DMEM (Gibco). All media were supplemented with 10% fetal bovine serum (HyClone; Thermo Fisher Scientific), 100 units/mL penicillin and 100 mg/mL streptomycin (both Gibco).

### 2.2. Reagents

The *IDH1*^MUT^ inhibitor AGI-5198 was purchased from MedChemExpress (Monmouth Junction, NJ, USA), *D*-2HG, oligomycin, olaparib, antimycin A, rotenone, carbonyl-cyanide-(trifluoromethoxy)phenylhydrazone (FCCP), L-glutamine and sodium pyruvate were purchased from MilliporeSigma (Burlington, MA, USA) and Sigma-Aldrich (St. Louis, MO, USA), and cisplatin was purchased from Pharmachemie B.V. (Haarlem, The Netherlands).

### 2.3. Cell Survival Analyses

Colony-forming assays evaluating cell survival were performed and analyzed as described previously [25]. From 5 to 500 cells/cm^2^ were seeded; higher cell densities were used with increasing treatment doses to obtain sufficient numbers of colonies. Prior to radiation exposure at doses of 0, 2 and 4 Gy (^137^Cs at a dose rate of approximately 0.5 Gy/min, at room temperature; Laboratory of Experimental Oncology and Radiobiology, Amsterdam UMC, location AMC), and exposure to cisplatin (0, 2.5 and 5 μM for 48 h), hyperthermia (42 °C for 1 h) and/or olaparib (10 μM for 48 h), cells were pretreated for 72 h with *D*-2HG, for 14 days with AGI-5198 or with solvent only (DMSO, 0.5%). Cells were treated for 1 h at 42 °C with hyperthermia in a thermostatically controlled water bath (Lauda aqualine AL12, Beun de Ronde, Abcoude, The Netherlands). Temperature was checked in parallel dishes, and the preferred temperature (±0.1 °C) was reached in circa 5 min with an air atmosphere of 5% CO_2_/95% and an air inflow of 2 L/min. Cells were treated with cisplatin 4 h after plating in the presence of 0–1 µM AGI-5198 or 0–10 mM *D*-2HG. After 10 days of treatment, a 0.05% crystal violet (Merck, Darmstadt, Germany) and 6% glutaraldehyde (Merck) mixture was used for fixation of cells for 2 h at room temp. Cell colonies were manually counted using a stereoscope (Leica MZ6; Leica Microsystems, Mannheim, Germany). As described previously, clones consisting of at least 50 cells were included and expressed as the clonogenic fraction; this is the number of colonies counted divided by the number of cells plated, corrected for the plating efficiency [26].

### 2.4. Cell Cycle Analyses

Cell cycle distribution was analyzed using the Click-iT^®^ EdU Imaging Kit (Invitrogen, Waltham, MA, USA), which allows for direct measurement of DNA synthesis. The thymidine analogue 5-ethynyl-2′deoxiuridine (EdU) was incorporated into S-phase cells and, as it had an Alexa Fluor^TM^ 647 picolyl azide attached, the fluorescence levels of individual cells reflected the cell cycle phase in which they were arrested. Cells (500,000 cells/mL) were plated in 6-well plates before treatment with hyperthermia (1 h at 42 °C). Subsequently, 16 h after treatment, 10 μM EdU was added for 1 h. Then, cells were harvested and fixed as suggested by the manufacturer’s protocol. Finally, the Click-it Plus reaction cocktail was added for 30 min (500 μL of phosphate-buffered saline (PBS), 10 μL of copper protectant, 2.5 μL of Alexa Fluor^TM^ 647 picolyl azide and 50 μL of Click-it^TM^ EdU buffer per plate). Cells were washed as suggested, and directly measured using a flow cytometer (BD FACS Canto II, 633/635 nm excitation filter; BD Biosciences, Franklin Lakes, NJ, USA).

### 2.5. γ-H2AX Immunofluorescence Staining and Quantification

Cells were plated on coverslips coated with 0.01% poly-*D*-lysine (Merck) 24 h prior to treatment. Different treatment combinations were used: cisplatin (5 mM for 24 h) alone or in combination with hyperthermia (42 °C for 1 h) 24 h prior to fixation, RT (1 Gy) and/or hyperthermia (42 °C for 1 h) for 30 min before fixation with 2% paraformaldehyde (Merck). After washing with PBS and permeabilization with TNBS (1% FCS + 0.1% Triton X-100 in PBS; MilliporeSigma) for 30 min, cells were incubated with a monoclonal mouse anti-γ-H2AX primary antibody (dilution 1:100 in TNBS) for 60 min. Next, the samples were washed with PBS and incubated for 30 min with a goat anti-mouse-Cy3 secondary antibody (dilution 1:100 in TNBS). After washing, DAPI-mounting gel (Vector Laboratories, Newark, CA, USA) was added and coverslips were sealed with nail polish. Directly afterwards, samples were imaged using a wide-field fluorescence microscope (Leica DM6 FS fixed-stage fluorescence microscope) [7]. The number of γ-H2AX foci per cell was analyzed as described previously [7]. Moreover, Cy3 and DAPI signals were captured using excitation/emission wavelengths of 550/570 nm for 400 ms and 360/460 nm for 50 ms, respectively. Stack images of at least 50 cells per sample were prepared. A custom-made software program was used to automatically score the number of γ-H2AX foci per cell.

### 2.6. Statistical Analysis

Data were processed and analyzed with Excel (Microsoft, Redmond, WA, USA) and visualized using Prism (GraphPad, La Jolla, CA, USA). Two-side tests were used with an α of 0.05. *p* values were calculated as described in figure legends; * *p* < 0.05; ** *p* < 0.01; *** *p* < 0.001; **** *p* < 0.0001.

## 3. Results

### 3.1. IDH1^MUT^ Cancer Cells Are Sensitive to Hyperthermia and This Induces Higher Sensitivity to RT and Chemotherapy

We first investigated the effect of hyperthermia as monotherapy using *IDH1*^MUT^ and *IDH1*^WT^ HCT116 colon cancer cells by performing colony-forming assays. Hyperthermia treatment alone caused a significantly higher reduction in the surviving fraction of 70% in *IDH1*^MUT^ cells compared to 40% in *IDH1*^WT^ HCT116 cells (Figure 3A). In addition, we evaluated the effect of hyperthermia in combined treatment with RT or chemotherapy on *IDH1*^MUT^ and *IDH1*^WT^ HCT116 cells. The combined treatment with hyperthermia (42 °C for 1 h) and RT using doses of 2 and 4 Gy increased the sensitivity of *IDH1*^MUT^ HCT116 cells dose dependently when compared with *IDH1*^WT^ HCT116 cells (Figure 3B). Combined treatment of hyperthermia and RT (2 and 4 Gy) increased the sensitivity of *IDH1*^WT^ HCT116 cancer cells by 51% and 85%, respectively, compared to hyperthermia treatment alone. In *IDH1*^MUT^ HCT116 cells, this combined treatment of hyperthermia and RT (2 and 4 Gy) increased the sensitivity by 76% and 85% compared to hyperthermia treatment alone, respectively. Compared to *IDH1*^WT^ HCT116 cells, *IDH1*^MUT^ HCT116 cells showed 60% (2 Gy) and 50% (4 Gy) more sensitivity to combined treatment of hyperthermia and RT. Hyperthermia is also known to improve the therapeutic effect of cisplatin [27,28], and since *IDH1*^MUT^ cancer cells are more sensitive to cisplatin, we investigated whether combining it with hyperthermia increased this sensitivity. As shown in Figure 3C, the combined treatment with hyperthermia and 2.5 or 5 μM cisplatin for 48 h increased sensitivity of *IDH1*^MUT^ HCT116 cells. Compared to *IDH1*^WT^ HCT116 cells, combination treatment of hyperthermia with 2.5 and 5 μM cisplatin showed a significant higher survival reduction of 65% and 93% in *IDH1*^MUT^ HCT116 cells, respectively. To exclude cell cycle distribution variations after hyperthermia treatment, we evaluated the effects of hyperthermia on the cell cycle of *IDH1*^MUT^ and *IDH1*^WT^ cancer cells. As illustrated in Figure 4, *IDH1*^MUT^ and *IDH1*^WT^ cancer cells responded similarly and went into cell cycle arrest (G2/M accumulation) after 1 h treatment with hyperthermia (42 °C), suggesting that the observed sensitivity after hyperthermia treatment did not have different effects on the cell cycle in *IDH1*^WT^ and *IDH1*^MUT^ cancer cells.

### 3.2. IDH1^MUT^ Inhibitor Protects lDH1^MUT^ Cancer Cells to Combination Treatment Induced by Hyperthermia

*IDH1*^MUT^ cancer cells are known to be sensitive to RT and cisplatin due to a decreased NADPH production capacity and increased metabolic vulnerability [26,29]. To confirm the causal relationship between *IDH1*^MUT^ and increased sensitivity to combination treatment with hyperthermia, we investigated whether the IDH1^MUT^ inhibitor AGI-5198 protects *IDH1*^MUT^ cells against combination treatment with hyperthermia. We exposed *IDH1*^MUT^ and *IDH1*^WT^ HCT116 cells to 1 µM AGI-5198 for 7 days before exposure to treatment with RT or cisplatin. AGI-5198 did not affect the sensitivity of *IDH1*^WT^ HCT116 cells, but it did reduce the sensitivity of *IDH1*^MUT^ HCT116 cells to combination treatment in a manner comparable to that of *IDH1*^WT^ HCT116 cells (Figure 3D–G). These data show that AGI-5198 reduced IDH1^MUT^-induced sensitivity to combination treatment of hyperthermia and RT or cisplatin.

### 3.3. Increased Numbers of DNA Double-Strand Breaks in IDH1^MUT^ Cancer Cells after Combination Treatment

The accumulation of DNA strand breaks, particularly double-strand breaks, is an important mediator of RT- and cisplatin-induced cell death in replicating cells [30]. Therefore, we investigated whether *IDH1*^MUT^ cells are sensitive to hyperthermia combined with RT or cisplatin due to a deficiency of the homologous recombination system leading to increased numbers of DNA double-strand breaks after treatment. Data presented in Figure 5 show a higher increase in the number of γ-H2AX foci in *IDH1*^MUT^ compared with *IDH1*^WT^ cells following exposure to RT (1 Gy) or cisplatin (5 μM for 24 h) and hyperthermia.

### 3.4. lDH1^MUT^ Cancer Cells Are Sensitive to PARPi and to Combination Treatment with RT

In addition to the combination of hyperthermia with RT and cisplatin, we investigated whether the combination with a PARPi increased the sensitivity of *IDH1*^MUT^ cancer cells. First, sensitivity to 10 μM olaparib for 48 h was assessed to confirm the causal relationship between *IDH1*^MUT^ and increased sensitivity to PARPi (Figure 6A). We then examined the combination treatment using olaparib and hyperthermia, which increased the sensitivity of *IDH1*^MUT^ HCT116 cancer cells compared to *IDH1*^WT^ HCT116 cancer cells. Furthermore, the response of *IDH1*^MUT^ cancer cells to the combination of olaparib, RT and hyperthermia was investigated. With the combination of the three modalities, a significant decrease in the survival of *IDH1*^MUT^ compared to *IDH1*^WT^ cancer cells was observed (Figure 6B). Reduced survival of *IDH1*^MUT^ cancer cells after hyperthermia treatment was also found in hyperthermia1080 chondrosarcoma cells with *IDH1*^MUT^ (Figure 6C).

## 4. Discussion

Chondrosarcoma, glioma and intrahepatic cholangiocarcinoma are types of cancer that often behave aggressively, often cannot be completely resected, tend to recur locally and commonly cause death through local progression. Since *IDH1*^MUT^ is causal in the development and progression of these types of cancer, many efforts have been made to discover specific vulnerabilities, especially in domains of metabolism and DNA damage induction and repair [31]. This resulted in a better understanding of the sensitization of cancer cells by *IDH1*^MUT^ to conventional chemotherapy and RT, but also to targeted agents, such as PARPis. In vitro and in vivo models demonstrated that *IDH1*^MUT^ cancers are sensitive to PARPis, and that this sensitivity to PARPis synergizes with temozolomide, RT or cisplatin treatment in vitro [6,8,9,10]. Ongoing clinical trials have been set up to test the effects of PARPis in *IDH1*^MUT^ solid cancers (NCT03212274), PARPis in recurrent *IDH*^MUT^ glioma (NCT03561870) and PARPis in combination with temozolomide in *IDH1*^MUT^ glioma (NCT03749187). A randomized phase 1/2 study of temozolomide in combination with the PARPi veliparib showed no benefit in recurrent temozolomide-refractory glioblastoma, but the *IDH1*^MUT^ status was not considered in the enrollment criteria or subgroup analyses [32]. This phase 1/2 study indicates that strategies and designs of clinical trials may be improved through a more profound understanding of the molecular mechanisms of these therapies.

Hyperthermia has been proven to be beneficial in the treatment of a number of cancer types and is generally applied in combination with RT and/or chemotherapy [17,21,22,23,33,34,35]. To our knowledge, combined treatment with a PARPi and hyperthermia has not been examined in the specific setting of *IDH1*^MUT^ tumors at either the experimental or clinical level. Our multimodality treatment approach of combining PARPis with hyperthermia to standard RT or cisplatin treatment is a novel strategy in *IDH1*^MUT^ solid cancers that can locally disable homologous recombination and sensitize cancer cells to DNA-damaging agents (Figure 7). Furthermore, we are the first to show that combined hyperthermia and PARPi in vitro increase the effectivity of RT and cisplatin treatment of *IDH1*^MUT^ and *IDH1*^WT^ cancer cells, with a 10-fold greater increase in *IDH1*^MUT^ than *IDH1*^WT^ cancer cells. We confirmed the causal relationship between *IDH1*^MUT^ and increased sensitivity to hyperthermia in combination with RT or cisplatin by pretreating *IDH1*^MUT^ cancer cells with an *IDH1*^MUT^ inhibitor (AGI-5198). *IDH1*^MUT^ inhibition reversed the sensitivity and protected *IDH1*^MUT^ cancer cells from the combination treatment of hyperthermia with RT or cisplatin.

Hyperthermia-induced sensitivity of *IDH1*^MUT^ cancer cells to combination treatment with PARPi is at least partly mediated by the reduced effectiveness of DNA repair systems in *IDH1*^MUT^ cancer cells. The relatively low redox status of *IDH1*^MUT^ cancer cells may also play a role. Hyperthermia is known to increase ROS levels and cause oxidative stress in cancer cells [15,18]; it also inhibits mitochondrial antioxidant systems via mechanisms such as reduced NADPH levels, which then contribute to increased intracellular ROS levels [18]. We and others have shown that the altered redox responses result in improved responses to therapy in *IDH1*^MUT^ cancers [26,36,37]. In the present study, we showed that *IDH1*^MUT^ cancer cells are more sensitive to hyperthermia than *IDH1*^WT^ cancer cells, and we envisage future research that will investigate hyperthermia-induced ROS formation and sensitivity mechanisms in *IDH1*^MUT^ cancer cells.

Another biological mechanism for hyperthermia that yields a strong enhancement of the effect of RT is inhibition of DNA damage repair [38], which has an impact on clinical results [39,40]. Locally applied hyperthermia also improves the effect of cisplatin [28] by enhancing cytotoxicity in the tumor without changing systemic toxicity [41,42]. Addition of hyperthermia to standard treatment regimens has shown favorable results for cervical carcinoma, soft tissue sarcoma, melanoma, rectal cancer and recurrent breast carcinoma in large randomized trials [17,33,35,43,44].

## 5. Conclusions

Hyperthermia may be considered, in view of our preclinical data, as a cornerstone of multimodality treatment regimes for patients with *IDH1*^MUT^ solid cancer. Regimens include (I) hyperthermia added to conventional treatment with RT and/or chemotherapy such as cisplatin and (II) hyperthermia in combination with PARPis. This multimodality treatment approach may be clinically achievable in certain settings, e.g., in case of an irresectable *IDH1*^MUT^ chondrosarcoma in an extremity and therefore deserves further study in a clinical trial.

## Figures and Tables

**Figure 1 cancers-14-06228-f001:**
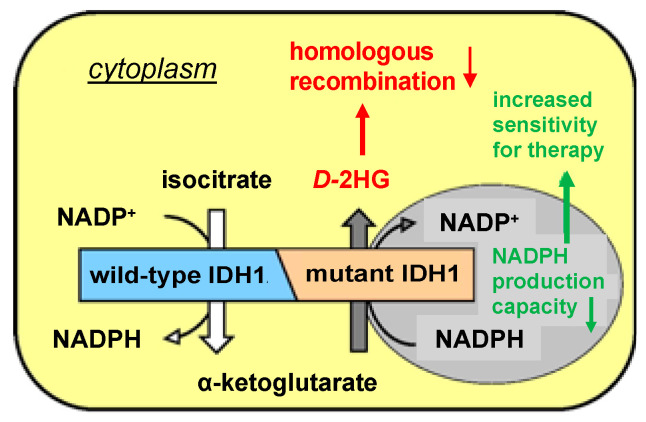
Schematic representation of the activity of an IDH1^WT/MUT^ heterodimer. The WT allele of the heterodimer produces αKG and NADPH, which are used by the mutated allele to produce D-2HG and NADP. The reduced NADPH production capacity leads to insufficient detoxification of, e.g., ROS, during irradiation and/or chemotherapy, whereas *D*-2HG inhibits homologous recombination, which causes increased therapeutic sensitivity.

**Figure 2 cancers-14-06228-f002:**
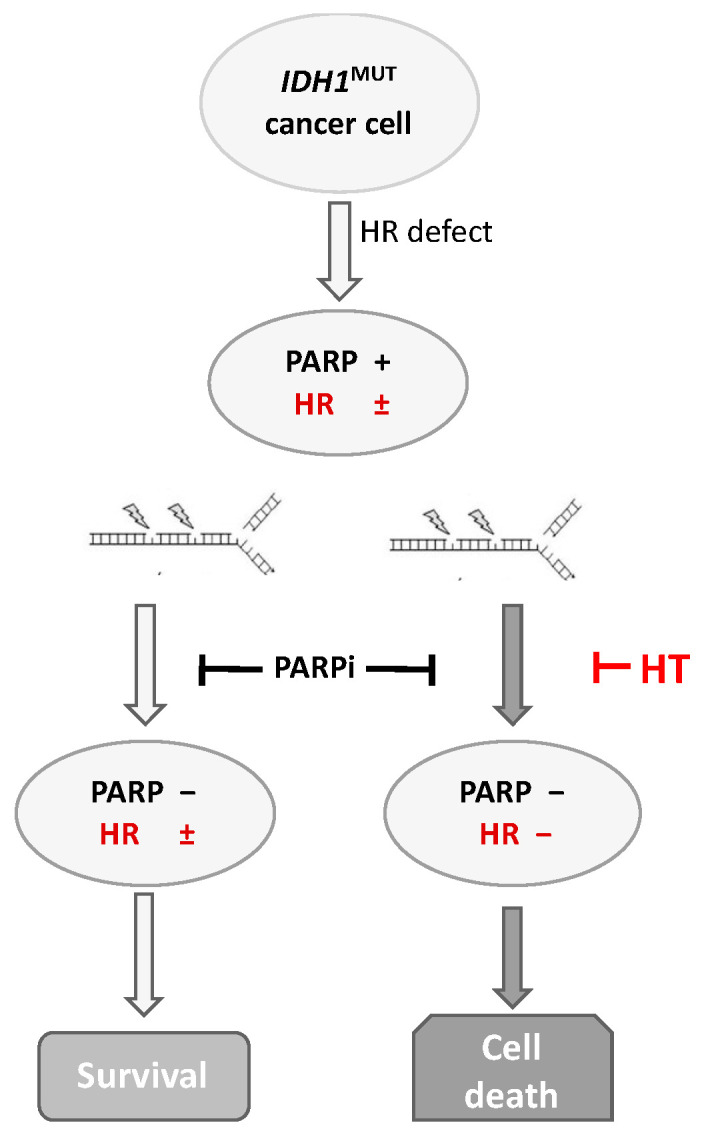
Schematic representation of the effect of hyperthermia and PARPi on the lethality of *IDH1*^MUT^ cancer cells. Accumulation of *D*-2HG has inhibitory effects on αKG-dependent di-oxygenases, which causes suppression of the homologous recombination repair system of double-strand breaks. Instead of monotreatment with PARPi, combination treatment with hyperthermia targeting the defect homologous recombination system in *IDH1*^MUT^ cancer cells leads to an increase in double-strand breaks and programmed cell death.

**Figure 3 cancers-14-06228-f003:**
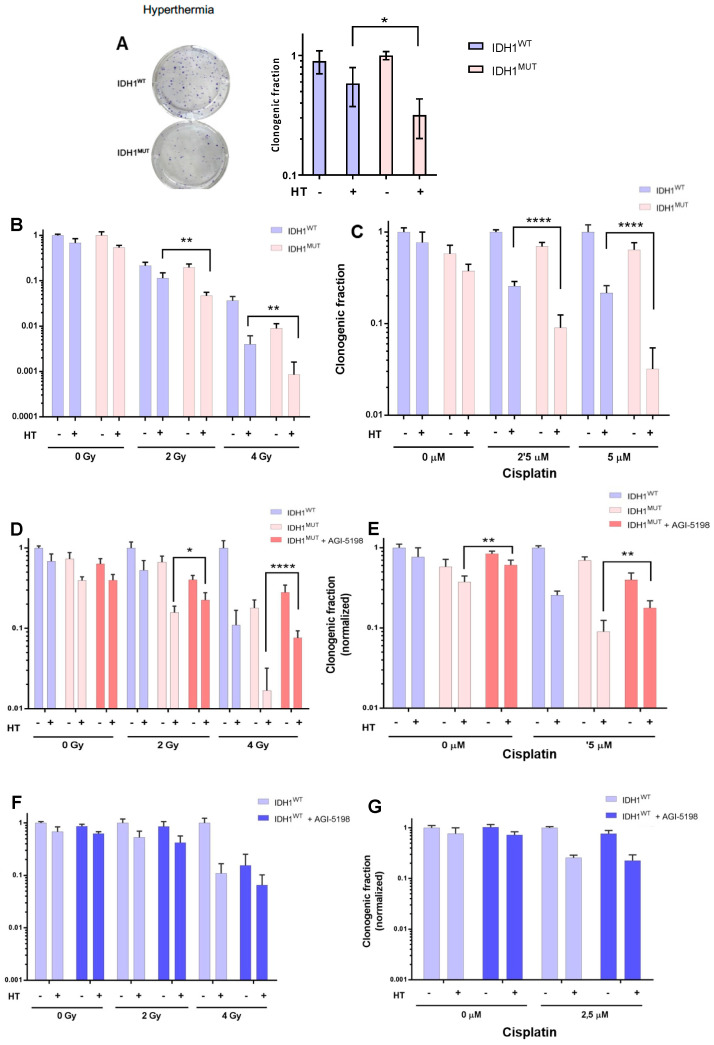
Clonogenic survival assays of HCT116 *IDH1*^WT^ and *IDH1*^MUT^ cells after combinations of anti-cancer treatment. (**A**) Images of crystal-violet-stained colonies (top *IDH1*^WT^, bottom *IDH1*^MUT^ cells) at day 14 after hyperthermia (42 °C for 1 h) and survival assay. Clonogenic survival assay (**B**) after combined RT (2 Gy and 4 Gy) and hyperthermia treatment of *IDH1*^WT^ T and *IDH1*^MUT^ HCT116 cells; (**C**) *IDH1*^WT^ and *IDH1*^MUT^ HCT116 cells after combined cisplatin (2.5–5 μM for 48 h) and hyperthermia (42 °C for 1 h) treatment; (**D**) *IDH1*^WT^ and *IDH1*^MUT^ HCT116 cells after (pre)treatment with AGI-5198 inhibitor in combination with RT and hyperthermia treatment; (**E**) *IDH1*^WT^ and *IDH1*^MUT^ HCT116 cells after (pre)treatment with AGI-5198 inhibitor in combination with cisplatin and hyperthermia treatment; (**F**) *IDH1*^WT^ HCT116 cells after (pre)treatment with AGI-5198 inhibitor in combination with RT and hyperthermia treatment; and (**G**) *IDH1*^WT^ HCT116 cells after (pre)treatment with AGI-5198 in combination with cisplatin and hyperthermia treatment. *p* values were determined using one-way ANOVA to reveal the difference between cisplatin-treated or irradiation-treated and untreated cells, using Tukey correction for multiple comparisons. Significance levels are shown by * *p* < 0.05; ** *p* < 0.01; **** *p* < 0.0001.

**Figure 4 cancers-14-06228-f004:**
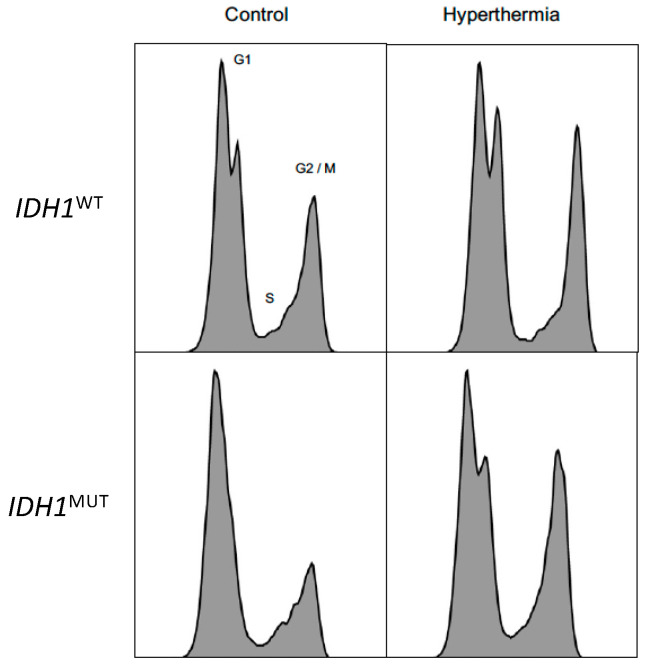
Cell cycle distribution patterns. Cell cycle distribution graphs of untreated (control) and hyperthermia treated *IDH1*^WT^ cells and *IDH1*^MUT^ HCT116 cells. *IDH1*^MUT^ and *IDH1*^WT^ HCT116 cancer cells responded similarly and went into cell cycle arrest (G2/M accumulation) after 1 h treatment with hyperthermia (42 °C).

**Figure 5 cancers-14-06228-f005:**
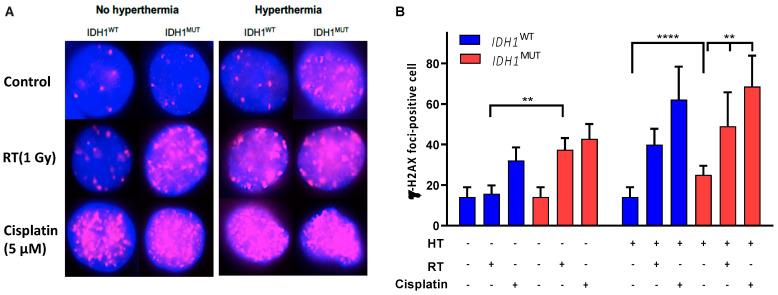
Double-strand breaks induced by hyperthermia, radiation (RT) and cisplatin in *IDH1*^WT^ and *IDH1*^MUT^ cancer cells. (**A**) Representative images of γ-H2AX foci in untreated cells (control), irradiated cells (1 Gy) and cells treated with cisplatin (5 μM for 24 h), with or without hyperthermia. γ-H2AX was stained immunocytochemically (red) to demonstrate DNA double-strand breaks and with DAPI (blue) to demonstrate DNA in nuclei. (**B**) Plots of γ-H2AX-positive foci per cell after cisplatin, RT and hyperthermia exposure of *IDH1*^WT^ and *IDH1*^MUT^ HCT116 cells. Plots are visualized with 95% confidence intervals and significance levels are shown by ** *p* < 0.01 and **** *p* < 0.0001. *p* values were determined using one-way ANOVA to reveal the difference between treated and untreated cells, using Tukey correction for multiple comparisons.

**Figure 6 cancers-14-06228-f006:**
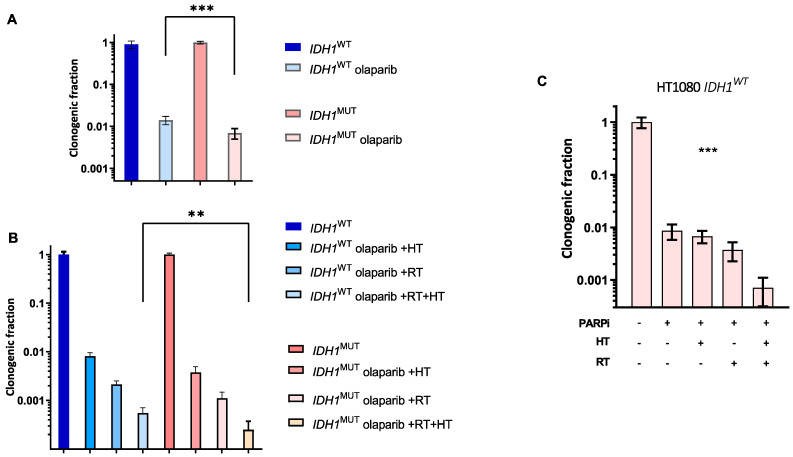
Clonogenic survival of *IDH1*^WT^ and *IDH1*^MUT^ cells after olaparib (PARPi) treatment (10 μM, 48 h) combined with hyperthermia (HT) and radiotherapy (RT). (**A**,**B**) *IDH1*^WT^ and *IDH1*^MUT^ HCT116 cancer cells and (**C**) hyperthermia1080 chondrosarcoma *IDH1*^MUT^ cells after indicated combination treatments. *p* values were determined using one-way ANOVA to reveal the difference between cisplatin-treated and -untreated cells, using Tukey correction for multiple comparisons. Significance levels are shown by ** *p* < 0.01 and *** *p* < 0.001.

**Figure 7 cancers-14-06228-f007:**
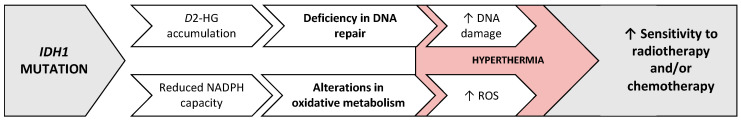
Effect of hyperthermia and PARPi in *IDH1*^MUT^ cancer cells. Hyperthermia-induced sensitivity of *IDH1*^MUT^ cancer cells is likely mediated by at least two components causing cell death: first, the reduced effectiveness of DNA repair systems, and second the relatively low redox status of *IDH1*^MUT^ cancer cells. A dysfunctional homologues recombination repair system and altered oxidative stress responses due to the altered metabolism explain the susceptibility of *IDH1*^MUT^ cancer cells to the combinational treatment with hyperthermia and/or PARPi.

## Data Availability

The data presented in this study are available on request from the corresponding author.

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
