# Peer review of "Hyperthermia as a Potential Cornerstone of Effective Multimodality Treatment with Radiotherapy, Cisplatin and PARP Inhibitor in IDH1-Mutated Cancer Cells"

_cancers, 2022, doi:10.3390/cancers14246228_

Round 1
Reviewer 1 Report
Originality is the priority in consideration of a paper publication. During the plagiarism check, I found the similarity rate was around 42% reported by the Turnitin system, which is the leading cause of MAJOR REVISION.

Author Response
Response to Reviewer #1
We appreciate the comments of the reviewer with respect to similarities found in the Material and Methods section and previous publications. These sections have been rewritten and relevant references are included.
Reviewer 2 Report
This work by Khurshed et al. evaluates the synergism between hyperthermia treatment and radiotherapy, cisplatin and PARPi in IDH1 cells. The article shows some interesting results, but many discussions regarding data are confusing and need to be eagerly improved. Therefore, it is my opinion that publication of the paper as it stands would be premature. Please address the comments below:
1) There are way too many abbreviations in the manuscript, which makes the text difficult to follow; there's usually no need to use an acronym (e.g., HT, RT, HR... or SSBs and DSBs).
2) Similarly, IDH1WT/WT and IDH1WT/MUT cells are not well defined; do authors mean ratios?
3) And the same applies to WT/R132H in Fig. 5.
4) Perhaps Figures 1 and 2 could be merged together.
5) The hyperthermia protocol is not defined (for example, did they heat cells in a water bath?). Apart from the whole-body hyperthermia, techniques usually refer to AC magnetic fields, laser, high-intensity focused ultrasounds, etc. Authors should comment on that.
6) On page 5, the sentence “Compared to HT treatment alone, combined treatment of HT and RT (2 and 4 Gy) increased the sensitivity of IDH1WT HCT116 cancer cells with 51% and 85%, respectively. In IDH1MUT HCT116 cells, combined HT and RT (2 and 4 Gy) treatment caused reduction of the surviving fraction with 76% and 85%” should be rephrased.
7) Survival should be also compared to control cells (i.e. IDH1 MUT and WT cells without treatment). I suggest including a Table.
8) Fig. 3 is missing, although there is a caption.
9) Authors should comment on differences in Fig. 4 IDH1 WT vs. MUT (control), for instance, the presence of 3 vs. 2 peaks.
Author Response
Response to Reviewer #2
We appreciate the comments of the reviewer and reviewed and revised the manuscript accordingly;
- The number of abbreviations (such as HT, HR, SSB and DSB) have been reduced and a list of Abbreviations has been added on page 10 of the manuscript.
- Schematic representation of the functions of IDH1MUT heterodimers is discussed in the Introduction and shown in Figure 1. We think this introduction and illustration is sufficient to understand the mechanism of the homodimer (IDH1WT) and heterodimer (IDH1MUT). We have added to the Introduction that IDH1WT/WT homodimer is indicated as IDH1WT and IDH1WT/MUT heterodimer is indicated as IDH1MUT, in accordance with the figures. Furthermore, IDH2 is not mentioned anymore as we only deal with IDH1 in this manuscript.
- We corrected figure 5 (IDH1WT/WT and > IDH1WT/MUT)
- Figure 1 introduces the IDH1 mutation and figure 2 represents the hypothesis and aim of the study. Merging the figures will confuse the reader.
- The hyperthermia protocol has been added to the Material and Methods section in more detail: “Cells were treated with hyperthermia by a thermostatically-controlled water bath for 1 h at 42 ºC. Temperature check was performed in parallel dishes and the desired temperature (±0.1ºC) was reached in approximately 5 min. Cells were heated in a 5% CO2/95% air atmosphere with an air inflow of 2 l/min.”
Regarding the comments requested by reviewer 2 of hyperthermia conditions in lab research versus clinical application. Indeed many techniques including AC magnetic fields, laser, high-intensity focused ultrasound are used in clinical treatment delivery, but these are not optimal for our study of hyperthermia on cells. The water bath provides the best control over treatment conditions and accurate temperature control for cell cultures, thus ensuring reproducible results. The details requested are relevant for clinical studies not for in vitro studies, therefore we did not include these details in the revised manuscript. - We rephrased the sentence into:
“Combined treatment of hyperthermia and RT (2 and 4 Gy) increased the sensitivity of IDH1WT HCT116 cancer cells with 51% and 85%, respectively, compared to hyperthermia treatment alone. In IDH1MUT HCT116 cells, this combined treatment of hyperthermia and RT (2 and 4 Gy) increased the sensitivity with 76% and 85% compared to hyperthermia treatment alone, respectively.”
7-9) Survival was indeed compared to control cells, but unfortunately figure 3 was missing in the version of the manuscript that was shared with the reviewers and this has been reported to the editors of Cancers. This figure illustrates survival assays of cells including control and combinations of anti-cancer treatment.
Reviewer 3 Report
The authors have uncovered the differential sensitivity of IDH1 mutant cancer cells to the sensitizing effect of hyperthermia to radiotherapy and chemotherapy. Mechanistically they show increased Double Strand Breaks in IDH1 mutant cells. In addition, they show that PARP inhibition further enhances the sensitivity of IDH1 mutant cancer cells in vitro.
Minor comments:
-The authors should better explain the rationale to add PARP inhibition to the combination of HT and RT/CT
-The Title is overstated; the authors provide only in vitro data, suggestion to alter into 'potential cornerstone'
Author Response
Response to Reviewer #3
- We appreciate the comments of the reviewer with respect to treatment with PARP inhibitors. As described in lines 67-73: “Homologous recombination deficiency also sensitizes IDH1MUT cancer cells to DNA repair-inhibiting agents such as poly-(adenosine 5′-diphosphate–ribose) polymerase inhibitors (PARPi) because of reduced repair of double strand breaks by homologous recombination deficiency and single-strand breaks due to PARP inhibition. Recent reports demonstrated sensitivity of IDH1MUT cancer cells to DNA damage and sensitization to PARPi in clinically relevant models, including patient-derived glioma and sarcoma cell lines and in vivo models.” We now also added literature on the rationale for the combination of PARP inhibition with HT and RT/CT in line 93-95: “The hyperthermia induced deficiency in homologous recombination DNA repair provided a strong rationale for adding PARP inhibition to the combination treatment of HT and RT and/or chemotherapy [21-23].” This model and the role of PARP inhibitors is illustrated in Figure 2.
- We agree with the comments of the reviewer with respect to the title of the manuscript, and inserted as suggested ‘potential cornerstone’ instead of “cornerstone”.
Round 2
Reviewer 1 Report
Originality is the priority in consideration of a paper publication. During the plagiarism check, I found the similarity rate is 35%, which is the leading cause of MAJOR REVISION.

Author Response
We appreciate the comments of the reviewer with respect to similarities found in the Material and Methods section and previous publications. These sections have been rewritten and relevant references are included.
Reviewer 2 Report
I am satisfied with the answers.
Author Response
We want to thank the reviewer for the effort to make the manuscript complete for the publication.